

# Regulation of HDAC11 gene expression in early myogenic differentiation

Qiao Li[1,2], Yan Z. Mach[1], Munerah Hamed[1], Saadia Khilji[1] and Jihong Chen[2]

[1] Department of Cellular and Molecular Medicine, Faculty of Medicine, University of Ottawa, Ottawa, Ontario, Canada
[2] Department of Pathology and Laboratory Medicine, Faculty of Medicine, University of Ottawa, Ottawa, Ontario, Canada

## ABSTRACT

Histone acetylation and deacetylation affect the patterns of gene expression in cellular differentiation, playing pivotal roles in tissue development and maintenance. For example, the intrinsic histone acetyltransferase activity of transcriptional coactivator p300 is especially required for the expression of myogenic regulatory factors including Myf5 and MyoD, and consequently for skeletal myogenesis. On the other hand, histone deacetylases (HDACs) remove the acetyl group from histones, which is critical for gene repression in stem cell fate transition. Through integrative omic analyses, we found that while some HDACs were differentially expressed at the early stage of skeletal myoblast differentiation, *Hdac11* gene expression was significantly enhanced by nuclear receptor signaling. In addition, p300 and MyoD control *Hdac11* expression in milieu of normal and signal-enhanced myoblast differentiation. Thus, HDAC11 may be essential to differential gene expression at the onset of myoblast differentiation.

## INTRODUCTION

In eukaryotic cells, gene expression is regulated by an integrated function between *cis*-regulatory DNA elements and the *trans*-acting transcription factors. Nevertheless, the contribution of *cis*-regulatory elements to gene expression depends on their accessibility for transcription factor binding in a chromatin environment, which is largely controlled by histone modifications (*Kouzarides, 2007*). As such, chromatin-modifying enzymes responsible for histone covalent modifications, often residue specific, are ultimately acting as transcriptional coactivators or corepressors (*Näär, Lemon & Tjian, 2001*). For example, histone acetylation is globally involved in transcriptional regulation, generally associated with active gene expression, and catalyzed by histone acetyltransferases (HATs) but reversed by histone deacetylases (HDACs) (*Wang et al., 2009*; *Lee et al., 1993*).

However, HATs and HDACs do not bind to DNA, to alter local and global chromatin state, DNA accessibility and gene expression, they must be recruited to the regulatory loci by DNA-binding transcription factors. Muscle regulatory factors (MRFs) including Myf5, MyoD, and myogenin, are class II basic helix-loop-helix transcription factors (*Murre et al.,*

Corresponding author
Qiao Li, qli@uottawa.ca

*1994*). MyoD is a master regulator of skeletal myogenesis (*Tapscott, 2005*) and its DNA binding generally leads to enhancer activation with marked H3K27 acetylation and muscle-specific gene expression (*Blum et al., 2012*; *Fong & Tapscott, 2013*). Therefore, MyoD binding in conjunction with HAT recruitment becomes an index of active myogenic enhancer (*Creyghton et al., 2010*). Genetic studies in mice and embryonic stem cells have conclusively demonstrated that the intrinsic HAT activity of transcriptional coactivator p300 is specifically required for the expression of Myf5 and MyoD, and consequently for skeletal myogenesis (*Polesskaya, 2001*; *Roth et al., 2003*). Through integrative omics studies, we have also recently defined myoblast chromatin states and residue-specific histone acetylation associated with p300 at the myogenic promoters and enhancers controlled by MyoD and myogenin in early myogenic differentiation (*Khilji et al., 2018*, *2020*).

HDACs are functional antagonists of HATs. They remove the acetyl groups from histones and are often associated with transcriptionally inactive loci and heterochromatin (*Wang et al., 2009*; *Vaquero et al., 2004*). In mammals, a total of 18 HDACs have been identified, categorized as the $Zn^{2+}$-dependent class I, II, IV, and the $NAD^+$-dependent class III HDACs (*Park & Kim, 2020*). They are essential for heterochromatin reorganization during terminal myoblast differentiation (*Terranova et al., 2005*), but also deacetylate non-histone proteins. HDAC1 deacetylates MyoD and silences MyoD-mediated gene expression (*Mal, 2001*; *Mal & Harter, 2003*). Snai1-HDAC1/2 repressive complex excludes MyoD from differentiation-specific regulatory elements, resulting in enhancer switching, a mechanism for regulating a distinct pattern of MyoD binding in the myoblasts and myotubes (*Soleimani et al., 2012*). In addition, HDAC4 impacts myogenic differentiation through its interaction with the transcription factor MEF2 and muscle homeostasis through deacetylation of myosin heavy chain (*Luo et al., 2019*; *Marroncelli et al., 2018*; *Lu et al., 2000*).

HDAC11 is the only member of the class IV HDACs, the most recently identified, and is localized to the nucleus (*Gao et al., 2002*). Interestingly, *Hdac11* gene expression increases in differentiating C2C12 and primary myoblasts (*Núñez-Álvarez et al., 2021*; *Byun et al., 2017*). Loss of HDAC11 upregulates cell cycle related genes and encourages muscle regeneration following muscle injury, although HDAC11 is expendable for muscle stem cell formation and adult muscle growth (*Núñez-Álvarez et al., 2021*; *Byun et al., 2017*). Nevertheless, the molecular mechanisms by which HDAC11 gene expression is governed remain unclear.

Bexarotene, a selective agonist for retinoic X receptor (RXR), has been used to treat cutaneous T-cell lymphoma for decades and been assessed for effectiveness for other pathologic conditions (*Gniadecki et al., 2007*; *Tanita et al., 2017*; *Mariani et al., 2017*; *Huuskonen et al., 2016*). We have previously established that bexarotene enhances myoblast differentiation through the function of RXR as a transcription factor to directly regulate MyoD gene expression (*AlSudais et al., 2016*; *Hamed et al., 2017*). We have also defined the functional mode of myogenic regulators in early myoblast differentiation using genome-wide chromatin state association (*Khilji et al., 2018*, *2020*; *Hamed et al., 2017*).

In this study, we explore the roles of p300 and MyoD in *Hadc11* gene expression in normal and signal-enhanced myoblast differentiation.

## MATERIALS AND METHODS

### Cell culture

C2C12 murine myoblasts (ATCC) were maintained in growth medium (GM), Dulbecco's Modified Eagle Medium (DMEM) supplemented with 10% fetal bovine serum (FBS), in the incubator at 37 °C with 5% $CO_2$. Myogenic differentiation was induced with 80% confluent myoblasts cultures using differentiation medium (DM), DMEM supplemented with 2% horse serum. Bexarotene was purchased from the LC Laboratories.

### Gene expression and histone acetylation profiles

The high degree correlation of the RNA-seq biological replicates have been described previously (*Li et al., 2022*; *Khilji et al., 2021*) with about 30 million mapped reads at 96% of mapping rate. The statistical power of the experimental design, calculated in *RNASeqPower* is 0.7. To determine differential gene expression between experimental conditions, a fold change greater than ±1.5-fold and below a false discover rate (FDR) of 5% were used to group the genes. The average enrichment of histone acetylation at the promoters of gene groups were visualized with ngs.plot (*Shen et al., 2014*). The ngs.plot calculates the coverage vectors for each query region based on specified alignment files, followed by normalization and transformation to generate an average profile with the number of reads normalized to the total number of mapped reads (in millions) in the dataset, in 20 bp bins within a 2 kb region, and centered at the transcription start sites (TSS). Integrative Genomics Viewer (IGV) was used for data browsing and representative snapshots. The heatmaps displaying differentially expressed genes were generated using the reshape2 and ggplot2 packages (*Wickham, 2007*, *2016*), based on the FPKM values (Fragments Per Kilobase of transcript per Million mapped reads) or log2-transformed *z*-scores.

### Reverse transcription qPCR analysis

As previously described, high-Capacity cDNA Reverse Transcription kit (Applied Biosystems, Waltham, MA, USA) was used for reverse transcription and SYBR® Green PCR Master Mix and HotStarTaq DNA polymerase (Qiagen, Hilden, Germany) for the quantitative PCR on a CFX96 Touch Real-Time PCR Detection System (BioRad, Hercules, CA, USA) (*Hamed et al., 2017*). Quantification of the targets, normalized to endogenous reference and relative to a calibrator control, was calculated using the formula $2^{-\Delta\Delta CT}$ as described (*Hamed et al., 2017*). *Tbp* primers for RT-qPCR have been described previously (*Hamed, Chen & Li, 2022*). *Hdac11* primers for analysis were as follows:

    *Hdac11*-F: 5′-TTACAACCGCCACATCTACC; R: 5′-GACATTCCTCTCCACCTTCTC.

### Quantitative ChIP analysis

C2C12 myoblasts were crosslinked and sonicated followed by chromatin immunoprecipitation as previously described (*Hamed et al., 2013*). Antibodies against
p300 and MyoD were purchased from Santa Cruz (sc-584x and sc-32758x). The immunoprecipitants were captured by Dynabeads protein-A, washed and eluted according to manufacturer's protocol (Invitrogen, Waltham, MA, USA) (*Hamed, Chen & Li, 2022*). Chromatin DNA was reverse crosslinked at 65 °C for overnight, purified with the DNA purification kit (Qiagen, Hilden, Germany) and amplified with SYBR® Green and HotStarTaq DNA polymerase (Qiagen, Hilden, Germany) on a CFX96 or CFX384 Touch Real-Time PCR Detection System (BioRad, Hercules, CA, USA) (*Hamed, Chen & Li, 2022*). Each sample was amplified in triplicate PCR reactions. Purified input DNA was used to generate a standard curve for the PCR amplification of each immunoprecipitation (*Hamed, Chen & Li, 2022*). The abundance of immunoprecipitated DNA was quantified as the percentage relative to the input chromatin DNA (*Hamed, Chen & Li, 2022*). Primer pairs used for the amplification were as follows:

*Hdac11*: F5′-GGGTGTAGGGGGAATGGAGA; R5′-TGCCTTGAACCTGTTTCCCT. Chromosome 15 gene desert, a negative control locus: F5′-TCCTCCCCATCTGTGTCATC; R5′-GGATCCATCACCATCAATAACC.

## Statistical analysis

All statistical analyses were performed using *R Core Team (2022)* or Microsoft Excel. Normally distributed data were analyzed by a two-tailed Student's *t*-test and data are presented as the mean ± SEM. Each experiment was repeated at least three times. Non-normally distributed data were analyzed by non-parametric two-sided Wilcoxon rank sum tests. A $p < 0.05$ was considered statistically significant. Statistical details and number of replicates are indicated in the figures.

# RESULTS

## Epigenetic feature associated with gene promoters in early myogenic differentiation

We have previously established a myoblast chromatin state model and defined residue-specific histone acetylation associated with active and poised enhancers in early myoblast differentiation (*Khilji et al., 2018*; *Hamed et al., 2017*). To survey the epigenetic features of myogenic promoters, we first profiled differential transcriptome which may contribute to myoblast fate decision by using our previously deposited RNA-seq data (GSE94560). Cuffdiff analysis discerned that 3,159 genes were significantly upregulated by over 1.5-fold in differentiating myoblasts in the first 24 h of differentiation, when compared to proliferating myoblasts, while 3,936 genes downregulated (Fig. 1A). As such, differential expression was observed on less than 30% of transcripts in differentiating myoblasts (Fig. 1B), revealing that a rapid change in myoblast fate transition arises from subsets of genes at onset of myoblast differentiation.

A previous study has reported that histone acetylation globally decreases at gene promoters during terminal myoblast differentiation (*Blum et al., 2012*). It is known that H3K9, H3K18, and H3K27 acetylation are found near transcription start sites (TSS), whereas H4K8 acetylation is associated to promoters and transcribed regions of active genes (*Li, Carey & Workman, 2007*; *Di Cerbo & Schneider, 2013*; *Wang et al., 2008*).

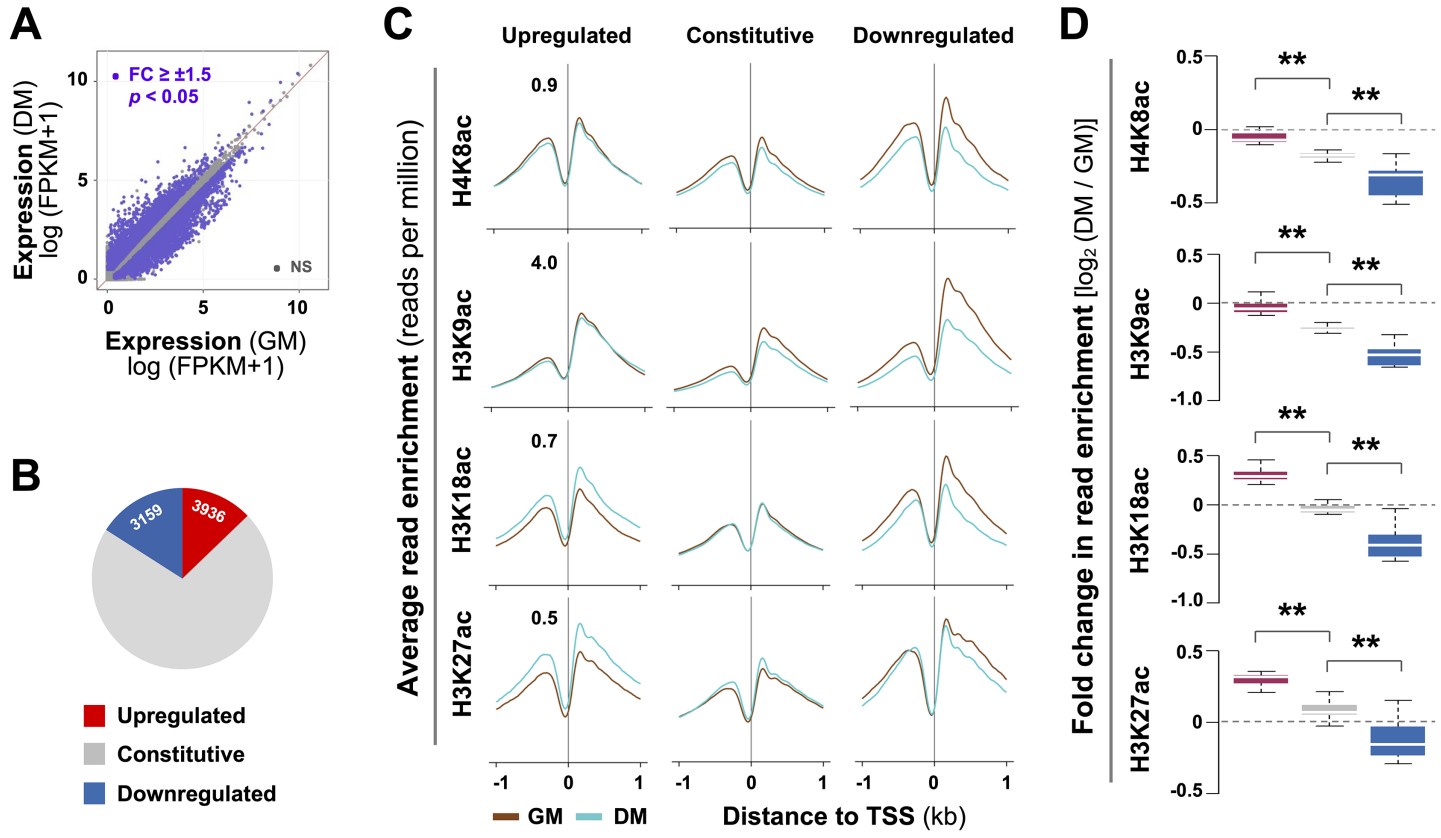

**Figure 1 Residue-specific histone acetylation in myogenic expression.** (A) C2C12 myoblasts were differentiated for 24 h (DM) and subjected to RNA-seq with proliferating myoblasts (GM) as controls. The expression of all annotated Ensembl genes was measured in fragments per kilobase per million mapped reads (FPKM). (B) Annotation of the gene groups based on significant change in expression ($\geq\pm1.5$ absolute fold change). (C) The average enrichment profiles of H4K8ac, H3K9ac, H3K18ac, and H3K27ac across the transcription start site (TSS, $\pm1$ kb) of genes that are upregulated, unchanged or downregulated in differentiating myoblasts compared to proliferating myoblasts. (D) Quantification is presented as $\log_2$-fold change in differentiating myoblasts when compared to proliferating myoblasts as in (C) with the same color key as for (B) (Wilcoxon rank sum test, $^{**}p < 2.2 \times 10^{-16}$).

We thus profiled residue-specific histone acetylation associated with promoters of differentially expressed genes in early myoblast differentiation by using our previously deposited histone acetylation ChIP-seq datasets (GSE94558) in matching conditions of the RNA-seq. As shown in Figs. 1C and 1D, at the promoters of downregulated genes, H4K8ac, H3K9ac and H3K18ac markedly decreased by 24 h of differentiation. Interestingly, change in H3K27ac signals was less evident, while the decrease in H3K9ac level was most pronounced (Figs. 1C and 1D). At the promoters of constitutively expressed genes, the levels of H4K8ac, H3K9ac, H3K18ac and H3K27ac were comparable between proliferating and differentiating myoblasts (Figs. 1C and 1D). However, at the promoters of upregulated genes, H3K18ac and H3K27ac levels were significantly augmented, whereas H4K8ac and H3K9ac signals remained similar in early differentiation (Figs. 1C and 1D). Taken together, our analysis indicates that changes in residue-specific histone acetylation at the promoters of differentially expressed genes may be intrinsic to myogenic activation at the onset of myoblast differentiation.

## Regulation of *Hdac11* by p300 in early myogenic differentiation

Histone deacetylases (HDACs) inhibit transcription by antagonizing HAT function through the removal of acetyl groups from the histones, which is essential for change of gene program in stem cell fate decision. Through RNA-seq analysis, we found that compared to other HDACs, the abundance of *Hdac11* mRNA was very low in proliferating myoblasts (Fig. 2A). While some HDACs were differentially expressed in early myoblast differentiation, the upregulation of *Hdac11* mRNA was most pronounced and significant, about 12-fold, in the first 24 h of differentiation compared to undifferentiated controls (Figs. 2A and 2B). To understand the control of HDAC11 function, we next sought to determine the molecular mechanisms by which *Hdac11* gene expression is governed in early myoblast differentiation.

Since p300 is a critical HAT required for skeletal myogenesis (*Polesskaya, 2001*; *Roth et al., 2003*), we examined p300 occupancy and concurrence of histone acetylation signature at the *Hdac11* locus following 24 h of myoblast differentiation, using our deposited p300 and histone acetylation ChIP-seq datasets (*Khilji et al., 2018*, *2020*). We also utilized our established model of myoblast chromatin states which is defined by large-scale analyses of co-occurrence of different histone marks (*Hamed et al., 2017*). As shown in Fig. 2C, the *Hdac11* promoter was classified as "active" in proliferating myoblasts based on the chromatin state model. Interestingly, the region spanning the active promoter was denoted as poised enhancer based on the presence of H3K4me1 signal but lack of H3K27ac (*Creyghton et al., 2010*; *Hamed et al., 2017*; *Rada-Iglesias et al., 2011*). At the *Hdac11* promoter, a moderate enrichment of H4K8ac, H3K9ac, H3K18ac signals in proliferating myoblasts was observed on the genome browser view of ChIP-seq read signals (Fig. 2C). However, at the poised enhancer, about 400 base pair upstream of the promoter, a distinct p300 occupancy as well as H3K27 acetylation was detected in myoblasts differentiated for 24 h (Fig. 2C). Moreover, the H4K8ac, H3K9ac, H3K18ac peaks were augmented, broadened, and overlapped with the p300 peak in the differentiating myoblasts (Fig. 2C).

Since MyoD is a master regulator of skeletal myogenesis (*Tapscott, 2005*), and can recruit the HAT p300 to instigate histone acetylation (*Puri et al., 1997*; *Hamed et al., 2013*; *Sartorelli et al., 1997*), we also examined the binding of MyoD to the *Hdac11* locus using a public available dataset (*Yue et al., 2014*). As shown in the genome browser, a prominent MyoD peak was found overlapping with the p300 read signal at the upstream of *Hdac11* promoter in myoblast differentiated for 24 h (Fig. 2C). Taken together, our bioinformatic analyses suggest that p300 and MyoD may be involved in the regulation of *Hdac11* gene expression in early myoblasts differentiation.

To exam the involvement of p300 and MyoD in the control of *Hdac11* gene expression, we designed a set of primers spanning the summits of p300 and MyoD read signals at the poised enhancer for quantitative ChIP analysis. As shown in Figs. 2D and 2E, p300 occupancy and MyoD binding at the *Hdac11* locus were both enriched by about 5-fold, following 24 h of differentiation as compared to proliferating myoblasts. On the other hand, the signals at a control locus were neglectable (Figs. 2D and 2E).
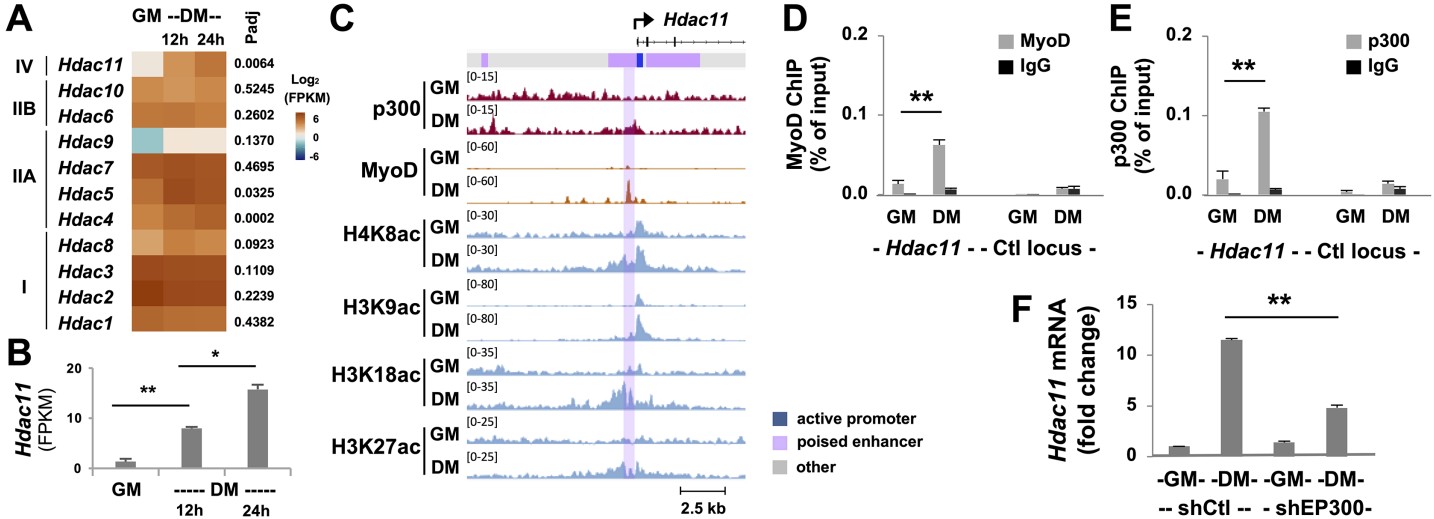

**Figure 2  Characterization of p300-associated *Hdac11* locus in differentiating myoblasts.** (A) Heat map of *Hdac11* read signals at 12 and 24 h of differentiation (DM) in log$_2$ FPKM (Fragments Per Kilobase of transcript per Million mapped reads) compared to proliferating myoblasts (GM). Shown are the adjusted *p* value (Padj) for DM 24 h *vs* GM. (B) Quantification of the *Hdac11* signals (*$p < 0.05$, **$p < 0.01$). (C) Genome browser view of p300, MyoD and histone acetylation signals at the *Hdac11* locus following 24 h of differentiation. Black bars show Refseq gene positions and ChromHMM track colors correspond to the color designated for each chromatin state. (D) Quantitative ChIP analysis for the *Hdac11* locus was performed using antibodies against p300; (E) Against MyoD with normal IgG antiserum as negative control. Quantification is presented as the percentage of ChIP enrichment in relation to the input DNA ($n = 3$). (F) *Hdac11* mRNA levels in the p300 knockdown myoblasts (shEP300) were analyzed using RT-qPCR analysis with nonsilencing shRNA (shCtl) as negative control. Quantification is presented as the fold changes relative to undifferentiated myoblasts, normalized to *Tbp* ($n = 3$).

To further determine the requirement of p300 for the control of *Hdac11* expression, we next used our established p300 shRNA knockdown myoblasts (*Chen et al., 2015*) to assess the levels of *Hdac11* mRNA in early myoblast differentiation. As shown in Fig. 2F, knockdown of endogenous p300 negatively impacted *Hdac11* gene expression, resulting in a significant reduction in *Hdac11* mRNA level in myoblasts differentiated for 24 h, when compared to corresponding nonsilencing shRNA control as determined by quantitative RT-PCR analysis. Taken together, our data suggest that p300 function is important for the upregulation of *Hdac11* expression at the onset of myogenic differentiation, possibly mediated by MyoD to activate the poised enhancer immediately upstream of the promoter.

## Effect of RXR signaling on *Hdac11* gene expression in early myoblast differentiation

RXR selective signaling enhances myogenic differentiation through a direct upregulation of MyoD gene expression as a transcription factor (*Hamed et al., 2017*), which may consequently impact *Hdac11* gene expression. As such, we profiled in detail the expression of HDACs using our deposited RNA-seq datasets on myoblasts differentiated in the presence or absence of an agonist of RXR, bexarotene. As shown in the heatmap, *Hdac11* mRNA was markedly upregulated as early as 12 h of differentiation (Figs. 2A and 3A). Interestingly, at 24 h, treatment with bexarotene significantly enhanced the expression of *Hdac11*, but not that of other HDACs (Fig. 3A). As shown in Fig. 3B, bexarotene augmented the level of *Hdac11* mRNA by additional three-fold, compared to untreated

myoblasts, determined by the quantitative RT-PCR analysis. Nevertheless, profiles of our deposited RXR ChIP-seq dataset (GSM2478303-2478305, at 24 h of C2C12 differentiation) did not exhibit specific RXR signal enrichment at the *Hdac11* locus. Therefore, the positive effect of RXR-selective signaling on *Hdac11* expression is possibly mediated through an indirect mechanism, directly enhancing MyoD gene expression (*Hamed et al., 2017*), which in turn positively regulates the *Hdac11* locus.

Since rexinoid-responsive myogenic expression is mediated through MyoD that is known to interact with HAT p300 (*Khilji et al., 2018*), we examined p300 and histone acetylation signature at the *Hdac11* locus in the context of RXR signaling. The Genome browser view of ChIP-seq read signals revealed that in differentiating myoblasts, the enrichment of p300, H4K8ac, H3K9ac, H3K18ac and H3K27ac read signals at the *Hdac11* promoter and putative enhancer were further augmented following 24 h of differentiation in the presence of bexarotene (Fig. 3C). Next, we used quantitative ChIP analysis to ascertain the involvement of p300 and MyoD in signal enhanced *Hdac11* gene expression. As shown in Figs. 3D and 3E, p300 occupancy and MyoD binding at the *Hdac11* locus were both further augmented by an additional 2-fold in response to bexarotene treatment when compared to untreated controls. Taken together, our data suggests that p300 and MyoD are involved in rexinoid-augmented *Hdac11* expression in early myogenic differentiation.

We also sought to determine the requirement of p300 function for *Hdac11* gene expression enhanced by RXR-selective signaling. As shown in Fig. 3F, p300 knockdown had a negative impact on RXR signaling-mediated *Hdac11* gene expression as exhibited by the attenuation of bexarotene-augmented *Hdac11* mRNA level following 24 h of differentiation, compared to untreated control, shown by the quantitative RT-PCR analysis. Taken together, our data suggest that p300 function is important for the upregulation of *Hdac11* gene expression, possibly through positive regulation of *Hdac11* locus in signal-enhanced myogenic differentiation.

## DISCUSSION

HDAC function is required for histone deacetylation to suppress gene expression which is especially crucial for regulating gene-specific cellular activities during stem cell fate decision. In this study, we have examined the regulation of *Hdac11* gene expression. We found that p300 HAT and muscle master regulator MyoD are intimately involved in augmenting *Hdac11* gene expression in both normal and signal-enhanced myoblast differentiation.

HDAC11 is expressed in the brain, heart, kidney as well as skeletal muscles (*Gao et al., 2002*), and it is upregulated during myoblasts differentiation (*Núñez-Álvarez et al., 2021*; *Byun et al., 2017*). We show that the mRNA level of *Hdac11* is considerably low in proliferating skeletal myoblasts, but markedly augmented in the first 24 h of differentiation to a comparable abundance of *Hdac1, Hdac2, Hdac3*, and *Hdac7* (Fig. 2A). In addition, these highly expressed HDACs in proliferating myoblasts are either downregulated to a certain degree or relatively constant during early differentiation (Fig. 2A). Thus, our data

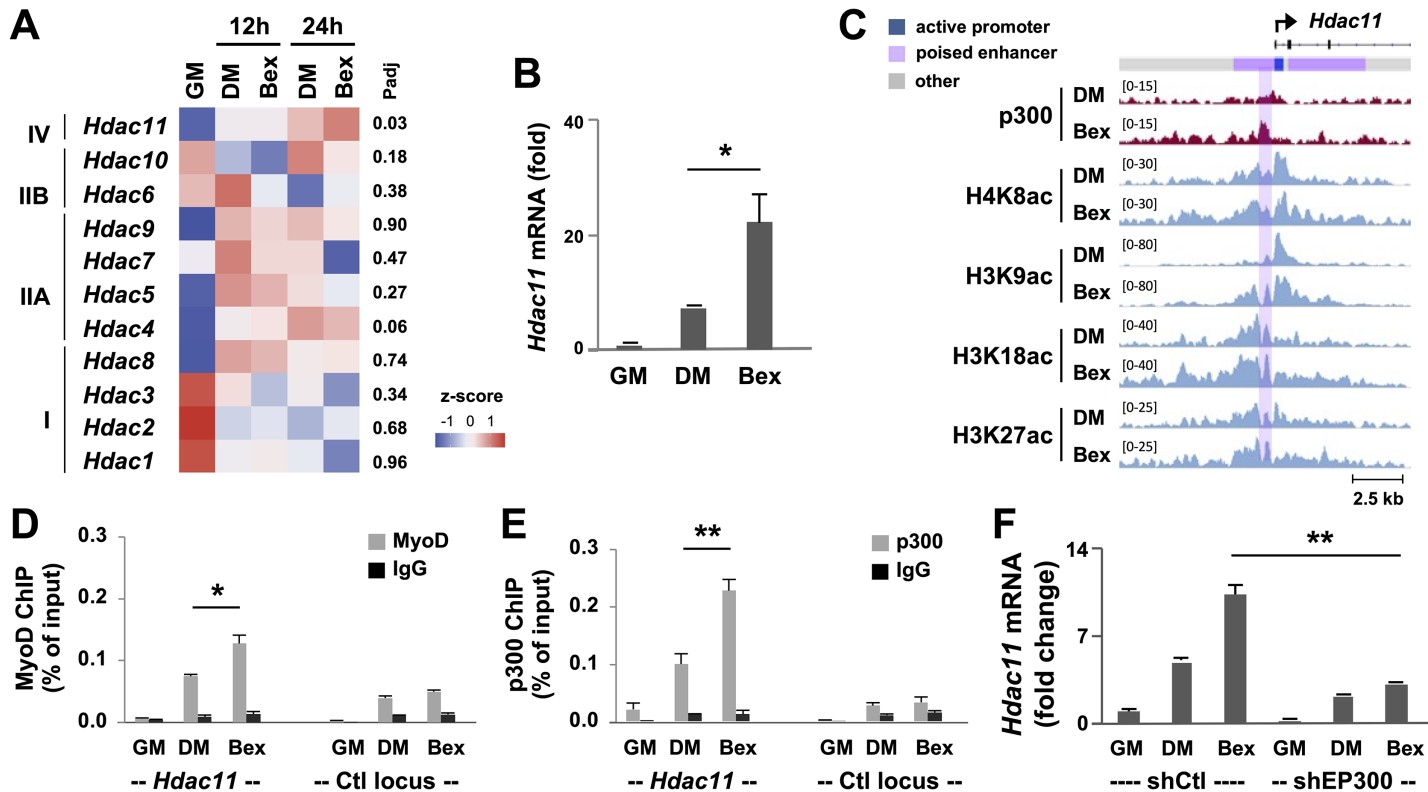

**Figure 3 RXR-selective signaling enhances *Hdac11* gene expression.** (A) C2C12 myoblasts were differentiated in the absence (DM) or presence of bexarotene (Bex, 50 nM) for 12- or 24-h, and subjected to RNA-seq analysis with proliferating myoblasts (GM) as controls. The heat map of bexarotene-enhanced *Hdac11* gene expression is shown along with other HDACs for comparison. Shown are the adjusted P value (Padj) for Bex 24 h *vs* DM 24 h. (B) Quantification of *Hdac11* mRNA by RT-qPCR analysis at 24 h of differentiation is presented as the fold changes in relation to proliferating myoblasts ($n = 3$, $^*p < 0.05$). (C) Genome browser view of p300 and indicated histone acetylation read signals at the *Hdac11* locus following 24 h of differentiation. Black bars show Ref-seq gene position and the ChromHMM track coded by the color designated to each chromatin state. (D) Quantitative ChIP analysis for the *Hdac11* locus was performed by using antibodies against p300; (E) Against MyoD with normal IgG antiserum as negative control. Quantification is presented as the percentage of enrichment relative to the input DNA ($n = 3$; $^{**}p < 0.01$). (F) *Hdac11* mRNA levels in the p300 knockdown myoblasts (shEP300) were analyzed by using RT-qPCR analysis with nonsilencing shRNA (shCtl) as negative control. Quantification is plotted as fold changes in relation to proliferating myoblasts after being normalized to *Tbp* ($n = 3$).

indicates that HDAC11 function may be particularly important for early myoblast differentiation.

Most interestingly, *Hdac11* is the only HDAC whose expression is significantly upregulated by RXR-selective signaling (Fig. 3). However, the augmentation of *Hdac11* gene expression only becomes evident at 12–24 h of differentiation compared to untreated controls, not prior (Fig. 3). We have previously reported that RXR-selective signaling directly upregulates gene expression of muscle-specific regulators including Akt2 and MyoD (*AlSudais et al., 2016*; *Hamed et al., 2017*). In addition, MyoD gene expression *per se* responds positively to RXR signaling in the first 12 h of myoblast differentiation (*Hamed et al., 2017*). Thus, the positive impact of RXR signaling on *Hdac11* gene expression may be a result of an indirect effect, mediated through the consequence of MyoD augmentation.

Based on our myoblast chromatin state model of proliferating myoblasts, the genomic region across the *Hdac11* promoter is classified as poised enhancer by the presence of

H3K4me1 but a lack of H3K27ac (*Hamed et al., 2017*). In skeletal myoblasts, poised enhancers are often associated with muscle specific genes, primed by MyoD and poised to be activated during myogenic differentiation (*Khilji et al., 2018*; *Hamed et al., 2017*). Here, we show that both MyoD and p300 HAT are associated to the *Hdac11* locus and the enrichment of H3K9ac, H3K18ac and H3K27ac signals is most pronounced in differentiating myoblasts and overlap with p300 and MyoD signal enrichments in normal and signal-enhanced differentiation (Figs. 2 and 3). This is consistent with our previous report that the coupling of H3K9ac, besides H3K18ac and H3K27ac, at the genome wide MyoD sites is especially associated with p300 occupancy in differentiating myoblasts (*Khilji et al., 2018*). Thus, HDAC11 may be responsible for histone deacetylation at target gene promoters to mediate differential gene expression at the onset of myoblast differentiation.

## CONCLUSION

Myogenesis is a multi-stage process including skeletal myoblast proliferation, differentiation, and fusion. At the onset of myoblast differentiation, chromatin-modifying enzymes are critical for cell cycle regulation to allow the activation of myogenic programs. For instance, *Hdac11* ablation affects cell cycle gene expression, leading to sustained myoblast proliferation albeit differentiation induction (*Núñez-Álvarez et al., 2021*).
In addition, p300 is critically required for MyoD gene expression, and essential to skeletal myogenesis (*Polesskaya, 2001*; *Roth et al., 2003*). Here, we found that p300 and MyoD are involved in the upregulation of HDAC11 in normal and signal enhanced myoblast differentiation. Thus, our data shed light on the interplay of chromatin modifying enzymes in cell cycle regulation to permit myogenic activation at the onset of myoblast differentiation.

## ACKNOWLEDGEMENTS

We thank our colleagues for a very supportive and collaborative research environment.

### Funding

This research was supported by an Operating Grant from the Natural Sciences and Engineering Research Council of Canada to QL (NSERC #250174). Munerah Hamed is a recipient of a scholarship from the Umm Al-Qura University. The funders had no role in study design, data collection and analysis, decision to publish, or preparation of the manuscript.

### Grant Disclosures

The following grant information was disclosed by the authors:
Natural Sciences and Engineering Research Council of Canada to QL: NSERC #250174.
Umm Al-Qura University.

## Competing Interests

The authors declare that they have no competing interests.

## Author Contributions

- Qiao Li conceived and designed the experiments, analyzed the data, prepared figures and/or tables, authored or reviewed drafts of the article, and approved the final draft.
- Yan Z. Mach performed the experiments, analyzed the data, prepared figures and/or tables, and approved the final draft.
- Munerah Hamed performed the experiments, analyzed the data, prepared figures and/or tables, and approved the final draft.
- Saadia Khilji analyzed the data, prepared figures and/or tables, and approved the final draft.
- Jihong Chen conceived and designed the experiments, analyzed the data, prepared figures and/or tables, authored or reviewed drafts of the article, and approved the final draft.

## Data Availability

ChIP-seq data are also available at GEO: GSE94558.

Omics data are available at the National Center for Biotechnology Information Gene Expression Omnibus (GEO): GSE94560.

The raw data for cell line experiments (Figures 2 & 3) are available in the Supplemental File.

## Supplemental Information

Supplemental information for this article can be found online at http://dx.doi.org/10.7717/peerj.15961#supplemental-information.

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
