# Peer review of "Regulation of HDAC11 gene expression in early myogenic differentiation"

_PeerJ, doi:10.7717/peerj.15961_

## Round 0.1 · original submission · Major Revisions

Dear authors, please carefully follow the reviewers' suggestions concerning your manuscript.

Reviewer 1 ·

Basic reporting

In their work, Li et al study the expression of the Hdac11 at two early stages of myogenic differentiation, as well as upon treatment with the Retinoid X receptor agonist. They explore its regulation through monitoring the presence of acetylation marks and myogenesis/acetylation-related proteins in the promoter region of the Hdac11 gene. The data is overall interesting and valuable for the field.
My major comment is that this study largely reutilizes the data already published by the same authors with a very similar message. The article is not self-sufficient enough: it does not have its proper logic, but is rather a rearrangement of the already published data with the logic determined by the datasets in hand and not the scientific question. It is not a systematic bioinformatics study either, as the authors are using only a very limited subset of the datasets available on myogenic differentiation, and mostly only those produced by themselves before. Thus, the article does not add on much to what is already published. Some other remarks are stated below.

Experimental design

Major remarks
lines 134-139: It would be convenient for the reader if the functions of H4K8Ac, H3K9Ac and H3K18Ac, less known than that of the H3K27Ac, were introduced before the description of the results. It is also not very clear from the results or discussion why they were explored in the first place.
lines 146-147: “our data suggests that reversible residue-specific histone acetylation at the gene promoters reflects the dynamics of HATs and HDACs in transcriptional regulation” - this conclusion does not stem from the data. No experiments concerning the reversible nature of acetylation or the actors (HATs and HDAC) of the aforementioned process were performed. Some of these might be already known, but the results presented in the paragraph neither support nor disprove these two statements.
line 162: “as well as our established chromatin state model” - please, include a couple of words describing the model.
line 168: “However, at the poised enhancer, upstream of the Hdac11 promoter, there were absent of read signals for p300 and histone acetylation in proliferating myoblasts (Fig. 2C).” - the signal is not enriched, but not absent according to figure 2C.
Figures 2DE, 3DE: The fact that the number of specific sequences in the DNA enriched for this sequence is less than 0.1% of the number of the same sequences in non-enriched DNA casts doubt on the validity of the experiment. Was the same total mass of DNA used for the input and the ChIP in the PCR reaction? If so, the ChIP was likely not successful. If not, the normalization to input is not justified. The authors should perform a control, where only the beads, but no antibodies are used for ChIP and compare the results with/without the antibodies to determine whether the ChIP is functional.
line 203: “at 24 hours, HDAC11 was the only rexinoid responsive gene among the HDACs (Fig. 3A)” - not the only one; according to the picture, the expression of HDAC 6 and 2 is also increased, while the expression of all other HDACs, except for the HDAC8, is decreased in response to Bex. In line 244, the authors state that HDAC11 was the only one significantly affected. Could the authors add multiple-comparison-adjusted p-values (padj) to the heatmaps at Figures 2 and 3?
Please, justify the exploration of the Retinoid X receptor (RXR) axis in the introduction, as there is currently no clear reason why it is explored other than the dataset availability.

Validity of the findings

-

Additional comments

Minor remarks
line 36: “are ultimately function” - language ambiguity; please, correct
line 44: “its binding to DNA generally corresponds to enhancer activation” - what is meant by “corresponds”?
line 52 “enhances” - language ambiguity; did you mean “enhancers”?
lines 68-70: “Nevertheless, the molecular mechanisms by which HDAC11 gene expression is governed remains unclear. In this study, we explore chromatin states and global differential gene expression underpinning myogenic differentiation.” - from this transition, it is not clear whether the study would explore the process of muscle differentiation globally or concentrate on the HDAC11 function in particular.
line 131: “As such, differential expression was observed on less than 30% of expressed genes in differentiating myoblasts ” - the RNA-seq reads represent the transcripts rather than genes, thus the percentage may be biased by alternative transcripts. Please, substitute the “genes” by transcripts, or specify that the transcripts coming from one gene were counted as one if this was the case.
line 152: “comparing other HDACs” - the preposition is missing, it is probably “compared to other HDACs”
line 168: “there were absent” - language
line 170: “a significant peak” - please, avoid using the term significant if not talking about statistical significance; if ‘statistically significant” was meant here, please, provide details on how this significance was determined.
line 197: “Retinoid X receptor (RXR) signaling enhances myogenic…” - according to the sentence structure, ‘enhances’ should be replaced by ‘enhancing’.
line 223: “We also sort to determine” - it is probably “sought to determine”
line 224: “enhanced by in RXR signaling”
line 236: “p300 and muscle master regulator MyoD are intimated involved” - language
line 246: the capital letter is missing in the beginning of the sentence

·

Basic reporting

Comments to the Authors

The authors addressed the molecular mechanism by which HDAC11 gene expression is upregulated in early muscle differentiation by comparing the chromatin states between proliferation and differentiation C2C12 cells at HDAC11 locus. The results showing the involvement of p300 and MyoD in the induction of HDAC11 expression at the onset of muscle differentiation, in normal and signal-enhanced (RXR) conditions, are clear and the figures are clear and well labelled and described. The paper structure and English language are correct.

However, there are several issues that need to be address and better discussed:

1- In the abstract, the authors refer to “Through integrative omic analyses, we found that while some HDACs were differentially expressed, at the early stage of myoblast differentiation, the upregulation of Hdac11 gene expression was most pronounced and significant”.
However, this is not a novel finding because it was already published in 2020, by Núñez-Álvarez et al. These authors reported that HDAC11 was the HDAC member whose expression changes the most at the transition from proliferating myoblasts to early differentiating myocytes (day 1 of differentiation) in satellite cell-derived primary myoblast (Figure 1A and B). The authors also reported the HDAC11 upregulation in human primary myoblasts upon differentiation and in differentiated C2C12 (Figures S1 and 1C, respectively). They also showed a very fast induction, being detected as early as 12 hours post differentiation (Figure 1C).

The abstract sentence should clarify that.

2- The introduction is fine, but in the last paragraph should include that it was previously published that HDAC11 was upregulated in C2C12 differentiating cells, as well as in human and mouse satellite cell-derived primary myoblast (Byun et al 2017 in Figures 1A and B; and Núñez-Álvarez et al.2020).

3- Regarding the methodology, the authors determine differential gene expression between proliferation and 24 hours of differentiation with a cut-off of  ±1.5 absolute fold change. The authors found 3.159 genes upregulated and 3.936 genes down regulated. However, they did not mention about which was the false discovery rate (FDR) used. FDR (the ratio of the number of false positive results to the number of total positive test results) allows to decide how many false positives they are willing to accept among all the results that can be called significant. It is important to mention which FDR has been used.


4- For MyoD and p300 ChIP, the authors do not indicate which is the control locus used and the corresponding primers in the Materials and Methods section.

5- In the discussion, the paragraph from lanes 238 to 243 should be rewritten, mentioning the two previous papers showing the up regulation of HDAC11 in early myoblast differentiation (Byun et al 2017; Núñez-Álvarez et al.2020). In addition, the authors should clarify that they compare mRNA levels, not protein levels, when they refer that HDAC11 rich the same level than HDAC1,2,3, and 7 upon differentiation.

6- In the discussion, the paragraph from lanes 244 to 251 is a bit confused. I think that the authors aim to comment about the fact that HDAC11 is upregulated by RXR signaling, but the RXR ChIP-Seq data do not exhibit specific RXR enrichment at the HDAC11 locus, suggesting an indirect effect. Please, revise this paragraph.

7- In the conclusion, the paragraph begins with “Adult muscle regeneration is a multi-stage process…..”

I would replace that for “Myogenesis is a multi-stage process….” , since the paper is not referring to muscle regeneration at any moment.

8- The last part of the conclusion should be modified, because is not focused on the paper results and because it is referred to the Núñez-Álvarez et al paper (reference 25) without being accurate.
“In addition, Hdac11 ablation affects the expression of genes involved in cell cycle progression, which leads to persistent myoblast proliferation irrespective differentiation induction (25). Therefore, HDAC11 likely plays an important role in histone deacetylation at these gene promoters to mediate cell cycle arrest to permit myogenic activation at the onset of myoblast differentiation”.

However, in this paper the authors reported that HDAC11-deficient satellite cells showed sustained proliferation, consistent with the deregulated expression of cell cycle genes observed in the RNA-Seq data and the observed delay in cell cycle exit in vitro and in vivo. However, the level of H3 acetylation marks in the analyzed promoter regions of some cycle-related genes, such as Aurka, Aurkb, and Pcna, did not show differences in shRNA-HDAC11cells. That would suggest that the upregulation of cell cycle-related genes is not mediated by increased H3 acetylation levels due to the reduction of HDAC11 histone deacetylase activity.

Experimental design

No new comment

Validity of the findings

No new comment

Additional comments

Minor comments

1- In lane 52, the word enhances I think it should be enhancers

2- In lane 202, when the authors refer to the HDAC11 upregulation as earlier as 12 hours of differentiation, I would also include Fig2A (Fig. 2A and 3A)

3- In Figure 3, it should be indicated DM or 24h of differentiation in the graphs corresponding to control and bexarotene treatment

4- In lane 206, please indicate in which conditions it was done the RXR ChIP-Seq. At 24 h of differentiation?

5- In figures 2A and 3A when it is indicated HDAC class, in HDAC11 class it is written Class V instead of Class IV (at least in my computer it is missed the number I)

6- In lane 218, it is written Figure 5D instead of Figure 3D

7- In Figures 2F and 3F better indicate in Y axis HDAC11 mRNA (fold change)

8- In lane 246, we should be in capital letter

9- In lane 267, it should be HDAC11 instead of HDAC1

---

## Round 0.2 · Minor Revisions

Please modify the manuscript as proposed by the Reviewer 1: "Could the authors add multiple-comparison- adjusted p-values (padj) to the heatmaps at Figures 2 and 3?"

Reviewer 1 ·

Basic reporting

The authors have satisfactory responded to most of the remarks, nevertheless one was omitted.
Could the authors kindly make the following modification from the previous review?

"Could the authors add multiple-comparison- adjusted p-values (padj) to the heatmaps at Figures 2 and 3?"

Experimental design

-

Validity of the findings

-

Additional comments

-

·

Basic reporting

The authors have made all the suggested changes/modifications in the manuscript.
I do not consider that the article needs any further modification.

Experimental design

No comment

Validity of the findings

No comment

Additional comments

Comments to the Authors

The authors have made all the suggested changes/modifications in the manuscript.
I do not consider that the article needs any further modification.

---

## Round 0.3 · accepted · Accept

The paper can now be accepted for publication.

Reviewer 1 ·

Basic reporting

-

Experimental design

-

Validity of the findings

-

Additional comments

-

·

Basic reporting

No comment

Experimental design

No comment

Validity of the findings

No comment

Additional comments

No comment